# Body Composition Evaluation and Clinical Markers of Cardiometabolic Risk in Patients with Phenylketonuria

**DOI:** 10.3390/nu15245133

**Published:** 2023-12-18

**Authors:** Luis M. Luengo-Pérez, Mercedes Fernández-Bueso, Ana Ambrojo, Marta Guijarro, Ana Cristina Ferreira, Luís Pereira-da-Silva, André Moreira-Rosário, Ana Faria, Conceição Calhau, Anne Daly, Anita MacDonald, Júlio César Rocha

**Affiliations:** 1Biomedical Sciences Department, University of Extremadura, 06008 Badajoz, Spain; 2Clinical Nutrition and Dietetics Unit, Badajoz University Hospital, 06008 Badajoz, Spain; mmercedes.fernandez@salud-juntaex.es (M.F.-B.); ana.ambrojo@salud-juntaex.es (A.A.); martamaria.guijarro@salud-juntaex.es (M.G.); 3Reference Centre of Inherited Metabolic Diseases, Centro Hospitalar Universitário de Lisboa Central, Rua Jacinta Marto, 1169-045 Lisboa, Portugal; anacristina.ferreira@chlc.min-saude.pt (A.C.F.); or rochajc@nms.unl.pt (J.C.R.); 4CHRC—Comprehensive Health Research Centre, Nutrition Group, NOVA Medical School, Universidade Nova de Lisboa, 1349-008 Lisboa, Portugal; l.pereira.silva@nms.unl.pt (L.P.-d.-S.); ana.faria@nms.unl.pt (A.F.); 5NOVA Medical School (NMS), Faculdade de Ciências Médicas (FCM), Universidade NOVA de Lisboa, Campo Mártires da Pátria 130, 1169-056 Lisboa, Portugal; andre.rosario@nms.unl.pt (A.M.-R.); ccalhau@nms.unl.pt (C.C.); 6CINTESIS@RISE, Nutrition and Metabolism, NOVA Medical School (NMS), Faculdade de Ciências Médicas, NMS, FCM, Universidade NOVA de Lisboa, Campo Mártires da Pátria 130, 1169-056 Lisboa, Portugal; 7Birmingham Children’s Hospital, Birmingham B4 6NH, UK; a.daly3@nhs.net (A.D.); anita.macdonald@nhs.net (A.M.)

**Keywords:** phenylketonuria, body composition, body fat, visceral fat, bioelectrical impedance, ultrasound, DXA scan, anthropometry, microbioma, metabolic syndrome

## Abstract

Cardiovascular diseases are the main cause of mortality worldwide. Patients with phenylketonuria (PKU) may be at increased cardiovascular risk. This review provides an overview of clinical and metabolic cardiovascular risk factors, explores the connections between body composition (including fat mass and ectopic fat) and cardiovascular risk, and examines various methods for evaluating body composition. It particularly focuses on nutritional ultrasound, given its emerging availability and practical utility in clinical settings. Possible causes of increased cardiometabolic risk in PKU are also explored, including an increased intake of carbohydrates, chronic exposure to amino acids, and characteristics of microbiota. It is important to evaluate cardiovascular risk factors and body composition in patients with PKU. We suggest systematic monitoring of body composition to develop nutritional management and hydration strategies to optimize performance within the limits of nutritional therapy.

## 1. Introduction

Phenylketonuria (PKU) is a rare autosomal recessive inherited metabolic disease that, when untreated, results in elevated phenylalanine (Phe) levels in the blood and central nervous system, as Phe crosses the blood–brain barrier and can cause irreversible neurological damage. The primary aims of treatment are to prevent or delay neurological complications as well as to provide optimal nutritional requirements for growth and development and a healthy body composition [1]. 

The main treatment is a low-natural-protein and Phe-restricted diet, supplemented with synthetic protein (protein substitute/medical food) in the form of Phe-free amino acids or low-Phe glycomacropeptide supplemented with amino acids [2]. Patients are dependent on special low-protein foods, such as bread and pasta, that are high in carbohydrates and may supply a significant proportion of their energy intake [3]. A subset of patients may be treated with pharmaceutical treatments such as sapropterin or pegvaliase [4].

Considering that an imbalanced low-protein, high-energy diet may increase the risk of overweight and obesity and that these both increase the risk of cardiovascular, metabolic, and other complications [5], the prevalence and susceptibility to excess adiposity in PKU is a matter of concern. This has led the industry to reformulate their protein substitutes to include less sugar and fewer calories [2].

Apart from this apparent diet-induced risk, overweight and obesity prevalence increases with age in the general population, and the first patients with classical PKU diagnosed by newborn screening are now in their fifties. Patients with treated PKU are expected to live their full life expectancy but may be at a higher risk of acquired metabolic disturbances, such as dyslipidaemia and insulin resistance (IR), which may raise the risk of cardiovascular events.

The majority of the literature only focuses on body mass index (BMI) interpretation, and a previous meta-analysis did not show a significant association between PKU and overweight [1]. Assessing body composition, mainly ectopic fat deposition, in patients with PKU is of great importance.

Metabolic syndrome (MetS) is a cluster of risk factors for atherosclerotic cardiovascular disease, which identifies individuals with an elevated risk of cardiovascular disease [6]. The most widely recognized risk factors are IR, atherogenic dyslipidaemia, high blood pressure (BP), and abdominal obesity [5,6,7].

The development of MetS is inherently linked to the metabolic activity of adipose tissue; that is, through adipose tissue inflammation, aberrant accumulation of ectopic lipid deposits, and mitochondrial dysfunction, particularly in skeletal muscle and liver. MetS is suggested to be a systemic manifestation of adipose tissue disease [8]. Adipose tissue secretes many factors, such as leptin and adiponectin, as well as proinflammatory tumor necrosis factor-alpha and interleukins. Inflammation leads to changes in visceral adipose tissue, including activated lipolysis, free fatty acid release, hypoxia, oxidative stress, and adipocyte apoptosis [9]. The main aim of this review is to discuss clinical and metabolic cardiovascular risk factors, explore the connections between body composition and cardiovascular risk, and examine various methods for evaluating body composition. Secondary objectives are to summarize scientific knowledge regarding cardiometabolic risk factors, techniques to evaluate body composition, association between body composition and cardiovascular risk, and causes of potential increased cardiovascular risk in patients with PKU.

## 2. Materials and Methods

This review aims to explore the relevant literature on cardiovascular risk factors, body composition assessment, potential underlying causes of increased cardiovascular risk in patients with PKU, and outcomes associated with body composition and metabolic complications in the context of PKU. The literature search was performed on PubMed without imposing any time constraints. The overall goal of this review was to ascertain the prevalence of cardiovascular risk factors among individuals with PKU.

The assessment of cardiometabolic risk in PKU was revised through a literature review process. A PubMed search using the query “PKU AND body composition ((phenylketonuria[MeSH Terms]) AND (body composition[MeSH Terms]))” yielded 20 results, while utilizing “All fields” instead of “MeSH Terms” yielded 50 results. By incorporating the “Humans” filter, a total of 38 results were obtained. Subsequent assessments, excluding animal studies and studies not related to body composition, reduced the count to 29. Additionally, 9 articles lacking body composition information were excluded. Despite exploring terms like “visceral fat”, “myosteatosis”, and “muscle AND fat infiltration”, no relevant results emerged.

Our pursuit of metabolic complications in PKU extended beyond studies centered on body composition. A supplementary search yielded 14 new articles (10 related to body composition and 4 regarding metabolic complications). The cumulative effort resulted in a total of 34 articles being considered for evaluation in the concluding review segment. This process is represented in Figure 1.

The topics regarding secondary objectives were revised following a similar process for the narrative review and included cardiometabolic risk factors, techniques to evaluate body composition, the association between body composition and cardiovascular risk, and causes of potential increased cardiovascular risk in patients with PKU.

## 3. Results

### 3.1. Classical Clinical Markers of Cardiometabolic Risk

Within the clinical markers, two groups are distinguished: anthropometric and metabolic markers.

#### 3.1.1. Anthropometric Markers

There are some anthropometrical measures which indicate cardiometabolic risk, but all of them have limitations. To evaluate **visceral fat**, anthropometric measurements are commonly used for their simplicity, speed, and accessibility [10].

**Body mass index** (BMI) assessment is common [11], and it classifies adult patients as overweight (25–29.9 kg/m^2^) or obese (≥30 kg/m^2^) [12], but it is not an appropriate measure of fat and skeletal muscle mass [13]. For a given BMI, the body fat percentage changes with age, depending on sex, ethnicity, and individual differences [14]. It underestimates the health burden from excess adiposity [15], and it is insensitive to the individual distribution of body fat [16]. BMI is related to total body fat but does not distinguish fat from muscle mass and should, therefore, be complemented by other anthropometric markers that evaluate the level of central adiposity since visceral fat is the parameter that has the best relationship with cardiovascular risk [17]. For these reasons, BMI should not be used for making clinically important decisions at the individual patient level [18], as it fails to identify all individuals with increased adiposity, which is underlined by the fact that some meta-analyses are unable to report linear associations between increased BMI categories and all causes of mortality [19].

**Waist circumference** (WC) is the classical anthropometric marker that best predicts intra-abdominal fat mass (FM) [20,21]. According to the National Institute of Health of the United States, a circumference > 102 cm in men and >88 cm in women is a higher risk [22]. WC can over- or underestimate the risk in adult patients with a higher or lower height percentile. Higher WC was associated with an increased risk of cardiovascular disease in women but not in men in one study [23] and was not associated with the blood fatty acid profile [24]. Currently, when assessing cardiometabolic risk with anthropometric measures, the use of WC together with BMI is proposed [25], though it does not accurately estimate cardiometabolic risk.

The **waist-to-hip ratio** (WHR) attempts to evaluate the relative distribution of fat; values < 0.8 in women and <1 in men are considered normal [26]. However, it is not usually helpful in the follow-up of patients because, in weight loss, both waist and hip circumferences can be reduced; therefore, the WHR does not change [27].

An attempt was made to improve WC by the **waist-to-height ratio** (WHtR) [26]; a value ≥ 0.5 could predict an elevated cardiometabolic risk more accurately than BMI or WC [28] and appears to be slightly more accurate in women than in men, although correlation indexes are not high [21]. Hwuang et al. proposed to use the WC index (WC/height^0.5^) combined with BMI, as it can play an important role when phenotyping adults for excess adiposity and associated health risks in research and clinical settings [29].

In 2010, in Italy, Amato et al. developed the **Visceral Adipose Index** (VAI) for a model of fat distribution. VAI considers anthropometric measures, such as BMI and WC, and lipid parameters, such as triglycerides and HDL. VAI was validated with magnetic resonance imaging (MRI), which found that VAI was independently associated with cardiovascular and cerebrovascular risk [30], and cut-off points were published [31]. Further studies have shown VAI to be positively associated with an increased risk of all-cause and cause-specific mortalities in a United Kingdom population [32] and correlated with visceral fat determined by ultrasound (US) imaging in an Indian population [33]. Nevertheless, VAI was only validated in certain clinical settings and only in a Caucasian population [34].

**Relative Fat Mass** (RFM) index was developed in 2018 by Woolcott and Bergman and validated with dual-energy X-ray absorptiometry (DXA) using data from the National Health and Nutrition Examination Survey (NHANES) [35]. It is useful to estimate changes in body and trunk fat by DXA and total abdominal fat by MRI and can be used as an estimate of whole-body fat, for which advanced imaging techniques are not feasible and are less useful in male patients [36].

Many other indexes obtained from anthropometric or mixed anthropometric plus biochemical parameters have been used in different populations to evaluate cardiometabolic risk [8,37,38,39,40]; however, some show wide individual errors and a lack of sensitivity in detecting change in body fat over time [41], and it is not yet fully clear which emergent anthropometric index may be best associated with cardiometabolic risk and if these vary according to sex and age [40].

Body FM is also estimated from **skinfold thickness**, according to several equations, such as by Durnin and Womersley [42], but these equations use the measurement of four to eight skinfolds and make an estimation of whole body fat mass percentage (%FM) [43] but not its distribution.

When assessing **muscle mass**, the presence of sarcopenia can be evaluated. Sarcopenia is defined as a loss of skeletal muscle mass, strength, and function, with loss of strength being the main defining parameter. The European Working Group on Sarcopenia in Older People (EWGSOP) [44] uses an algorithm for detection based on three parameters: muscle mass, walking speed, and muscle strength. Anthropometric measurements are available to estimate reduced muscle mass (in addition to more specific techniques): calf circumference (cut-off point < 31 cm) and mid-arm muscle circumference (cut-off point < 12.5 cm). To assess gait speed, the Up and Go test may be used. To assess loss of muscle function, hand dynamometry is performed. Depending on the diagnostic criteria, the prevalence of sarcopenia varies widely [45]. Anthropometry methods are usually more imprecise, and the combined use of calf circumference, fat-free mass index, and appendicular skeletal muscle index increases the sensitivity for the diagnosis of sarcopenia associated with malnutrition [46]. 

This requires more specific methods to assess body composition than being used at present: computed tomography (CT), MRI, DXA, bioelectrical impedance analysis (BIA), and nutritional US [47,48].

The advantages and limitations of anthropometry and other techniques are shown in Appendix A.

#### 3.1.2. Metabolic Cardiovascular Risk Factors

Classical metabolic risk factors of cardiovascular disease are hyperglycemia/IR, hypertriglyceridemia, low HDL [5], and low-density lipoprotein (LDL).

Other markers of cardiometabolic risk are TNF-alpha, C-reactive protein (CRP), interleukin (IL)-6, and plasminogen activator inhibitor (PAI)-1. 

Increased secretion of TNF-alpha leads to increased aggregation of activated bone marrow macrophages and macrophage infiltration of adipose tissue. Cellular oxidative stress within the mitochondria, nucleus, and endoplasmic reticulum due to excessive fuel supply leads to mitochondrial dysfunction [49], and increased visceral adipose tissue is associated with the development of type 2 diabetes mellitus (DM2). Adipose TNF-alpha expression is directly related to the development of IR in obesity [50].

CRP from liver synthesis plays an important role in the inflammatory processes associated with MetS, insulin sensitivity, endothelial dysfunction, and fibrinolysis failure [51].

MetS has been recognized as a proinflammatory, prothrombotic state associated with elevated levels of CRP, IL-6, and PAI-1 [52]. Inflammatory and prothrombotic markers are associated with IRs and an increased risk for subsequent cardiovascular disease and DM2 [53].

### 3.2. Body Composition and Cardiometabolic Risk

#### 3.2.1. Types of Adipose Tissue and Cardiometabolic Risk

Excessive body weight or total adiposity, in addition to body fat distribution, may determine metabolic risk [54]. BMI and total adiposity only correlate with cardiovascular risk at a population level, but fat distribution and adipose tissue dysfunction correlate better at an individual level. Adipocyte dysfunction includes impaired adipose tissue expandability, adipocyte hypertrophy, altered lipid metabolism, and local inflammation [55].

There are two main types of adipose tissue: brown (BAT) and white (WAT). WAT stores energy as fat and releases hormones and cytokines that modulate metabolism and insulin sensitivity/resistance. BAT, on the contrary, maintains body temperature by heat production via thermogenesis in its numerous mitochondria, mediated by uncoupling protein 1 (UCP1), which is tissue-specific [56]. There are two other types of adipose tissue: beige adipocytes arise from WAT by increasing their number of mitochondria and then initiating thermogenesis, and pink adipocytes can be seen in mammary tissue during lactation [57].

The location of adipose tissue is also important as subcutaneous adipose tissue (SAT) stores the excess of triglycerides, and when adipocytes in SAT reach their maximum capacity to store fat, this leads to ectopic fat accumulation. This fat deposits preperitoneally around and within organs without adipose tissue (skeletal muscle, liver, pancreas) and is known as visceral adipose tissue (VAT), which may lead to impairment of the functions of surrounding tissues and organs. Thus, abdominal VAT, epicardial fat, fatty liver, and myosteatosis, the fatty infiltration of skeletal muscle (see below), are known to be associated with IR, DM2 and, thus, increase cardiovascular risk [58,59,60].

The results of different epidemiological studies conducted over the last 30 years have shown that VAT is an independent risk marker of cardiovascular and metabolic morbidity and mortality. Evidence also suggests that ectopic fat deposition, including hepatic and epicardial fat, could contribute to increased atherosclerosis and cardiometabolic risk [59]. In addition to the loss of muscle mass and strength, aging also leads to the redistribution of adipose tissue, through which subcutaneous adipose tissue relocates to more detrimental locations [60].

Ectopic fat deposition can enhance atherosclerosis and endothelial dysfunction by impaired adipogenesis, adipokine dysregulation, inflammation, increased circulating free fatty acids, oxidative stress, adipose tissue hypoxia, and local and systemic lipotoxicity. Ectopic fat can also increase the risk of cardiometabolic disease by modulation of risk factors such as hypertension, diabetes, and dyslipidemia. Nevertheless, there is great individual variability in body fat distribution for a given BMI, as susceptibility to store fat and where to store it is, in part, determined by genetics, and the body’s response to excess energy accumulation is not uniform [61].

#### 3.2.2. Sarcopenia, Myosteatosis, and Cardiometabolic Risk

Myosteatosis, as an increase in intra- and intermuscular fat [62], and sarcopenia (see Section 3.1.2) are recognized as the muscular expression of MetS [63,64,65]. However, they remain underestimated and underdiagnosed.

Sarcopenia has a multifactorial origin, from aging to chronic systemic inflammation, but it is important to highlight the MetS in skeletal muscle alterations. IR and the imbalance that occurs in the insulin-sensitive tissues, mainly the liver, the muscular system, and the hypothalamus–hypophysis axis, should be highlighted [66].

There is an association between sarcopenia and liver fibrosis in non-alcoholic fatty liver disease (NAFLD). In a state of IR, NAFLD can develop as a consequence of an increased concentration of free fatty acids (FFAs) in the liver. This higher liver FFA is related to an increased de novo lipogenesis produced by hyperinsulinemia. Furthermore, increased hepatic gluconeogenesis results in a persistent catabolic state of the muscular system, with the aim of supplying the liver with protein-derived amino acids as a substrate for glucose synthesis, thus exacerbating sarcopenia [66,67]. Because of its association with MetS, some authors propose changing the name NAFLD to metabolic dysfunction-associated fatty liver disease (also referred to as MAFLD) [67].

In addition, sarcopenia interferes at the level of the hypothalamus–hypophysis axis and growth hormone (GH) [59]. GH has a muscle trophic effect by enhancing protein synthesis and beta-oxidation of FFA. Obesity is related to reduced GH activity, decreasing muscle mass in a way potentially reversible with weight loss [66].

Sarcopenia is difficult to define in people with limited mobility. In these patients, sarcopenia is not only due to loss of muscle mass and concomitant disability but also to histological modifications such as changes in muscle fibers. The concept of sarcopenia is therefore extended to include both quantitative and qualitative changes in muscle tissue. In disuse atrophy, muscle changes are due to a reduction in muscle protein synthesis. In contrast, after injury to the central nervous system, muscle changes are mainly due to increased proteolysis that appears at the onset of the disease. A rapid treatment with nutritional supplementation and exercise, among others, is important to probably avoid muscle modifications and atrophy in neurological patients [68].

Age-associated obesity and muscle atrophy are intimately connected and are reciprocally regulated. During aging, and independently of changes in body weight, adipose inflammation leads to the redistribution of fat inside the abdomen and fatty infiltration in skeletal muscles, resulting in decreased overall strength and functionality, affecting mitochondria and producing proinflammatory cytokines that exacerbate adipose tissue atrophy and chronic inflammation, establishing a vicious cycle that promotes sarcopenic obesity [69].

There are two possible fat depots within skeletal muscle, either within the myocytes (intramyocellular fat) or visible fat within the fascia surrounding skeletal muscle (intermuscular fat) [69].

In people with DM2, a lack of myocyte response to an insulin stimulus has been observed and, consequently, impaired glucose uptake in muscle cells, with skeletal muscle being a crucial organ for maintaining glucose homeostasis [69].

Several studies showed that decreased muscle mass is a risk factor for coronary atherosclerosis [70], and good skeletal muscle quality itself is an important protective factor for subclinical coronary atherosclerosis [71].

The Coronary Artery Risk Development in Young Adults (CARDIA) study showed that higher intermuscular adipose volume measured by abdominal CT was significantly associated with coronary artery calcification [72].

Finally, public health messaging should focus on visceral and ectopic fat in addition to excess body weight to better combat the growing epidemic of obesity [73].

### 3.3. Techniques to Evaluate Quantity and Quality of Adipose and Muscle Tissues

Body composition can be assessed at several levels depending on the clinical concerns. Methods used are classified as direct, criterion, and indirect. Direct methods can analyze from the atomic through to the cellular levels, and those used for clinical purposes include isotope dilution and total body counting [74]. Criterion methods describe amounts and distributions of tissues or measure their densities and include air-displacement plethysmography (ADP), MRI, CT, US, and DXA. Indirect methods provide estimates or indices of body composition and are based on measurements from direct or criterion methods, and the most frequently used are anthropometry and BIA. Indirect methods have larger predictive errors than direct methods since they depend on biological interrelationships among measured body components and tissues and are affected by clinical conditions [74].

Data obtained from different body composition methods are not always interchangeable. Using a four-compartment model based on DXA, isotope dilution, body density, BIA, and skinfold thicknesses, age- and sex-specific reference data were determined for individuals aged 5–20 years, with correlations between SD scores of ≥0.68 for adiposity and ≥0.80 for fat-free mass (FFM) [75].

Bicompartmental methods, in particular, measure FM and FFM and are preferred in clinical settings. Using a bicompartmental model, it should be considered that fatness is more accurately estimated than leanness because FM has a more constant density (0.9007 g/mL) than FFM, which is dependent on its different water content [69]. In addition, FM is more homogenous, comprising predominantly adipose tissue, while FFM is a complex compartment containing not only skeletal muscle but also bone, organs, and blood [76].

#### 3.3.1. Dual-Energy X-ray Absorptiometry

Dual-energy X-ray absorptiometry was primarily designed to measure bone density and is the International Society for Clinical Densitometry’s (ISCD) recommended method for the assessment of real bone mineral density [77,78]. DXA is a two-dimensional imaging technique, with images being generated using X-rays at two different energy levels (high and low), which differentiate between bone and soft tissue. This generates a three-compartment model measuring FM, FFM, and bone mineral content [79,80]. Measurements can be divided regionally for the head, trunk, arms, and legs in addition to whole-body scans. DXA has been found to be more accurate than density-based methods for estimating total body fat, although no method is without some limitations [81]. The technique assumes that FFM has a constant composition of water, protein, and minerals, although these factors vary with age, gender, and disease [82]. Inaccuracies have been reported with older people when assessing obesity and in some diseases [83]. Limitations vary with body shape and trunk composition. It involves prediction rather than measurement, and soft tissue estimation in this area is less accurate than those measured in limbs. However, DXA is quick and uses low ionising radiation; it is a relatively low-cost technology and is accurate and easy to perform given skilled operators. DXA generates both bone and soft tissue data. With the exception of pregnancy, DXA is safe, although it is recommended only to be performed a maximum of twice a year, which is exposure comparable to that of an intercontinental flight [84].

DXA has a range of clinical applications, including assessment of associations between adipose and lean mass, and is used to provide information on truncal FM, which can predict disease risk. It can provide estimates of visceral fat using validated predictive algorithms [85]. Absolute measures of FFM provide an important indicator of total body protein status and are associated with physical strength outcomes. DXA measures excess adiposity with more accuracy than BMI and quantifies total fat and lean soft tissue in addition to truncal and visceral fat, which is helpful in the evaluation of cardiometabolic risk. DXA also provides functional data on both lean mass and bone mineral content. The strongest significant predictor of bone mineral content is lean body mass, and deviations from this close relationship can identify subjects with a chronic bone and or muscle defect [86].

New strategies such as half body assessments allow accurate body scans and analysis of individuals over the scan field limits. Although DXA is a projective imagining technique, new methods have allowed the differentiation of subcutaneous and intra-abdominal visceral fat. Visceral fat is strongly associated with cardiovascular risk, while lean mass is the site of insulin-mediated glucose uptake, determining total body insulin sensitivity [87,88]. Changes in lean mass during childhood could influence long-term morbidities, highlighting the importance of sustained monitoring.

The advantages and limitations of DXA and other techniques are shown in Appendix A.

#### 3.3.2. Bioelectrical Impedance Analysis

There are several BIA devices, single-frequency (usually 50 kHz) or multifrequency, bipolar or tetrapolar, whole-body or segmental, and vectorial, with low variability (1–2%) in repeated measures. The precision and accuracy are influenced by many factors related to the patient, environment, protocols, instrumentation, and the assumptions taken to make the prediction, as this technique does not measure body composition but rather estimates it from the different resistance measurements of different tissues to a low-intensity (500–800 mA), high-frequency electric current using a bicompartmental model of lean and FM. BIA also measures reactance, impedance, and phase angle to assess total body water and, by assuming that FFM hydration is constant (73%) and contains virtually all the water and electricity-conducting electrolytes, predicts fat-free mass. Therefore, changes in hydration are the main limitations of the technique, which tend to overestimate FFM and underestimate FM in children who are more hydrated, as well as adult individuals with BMIs below 18 or above 34 kg/m^2^ [47,89].

BIA has a 3% to 5% (coefficient of variation) accuracy when compared with other techniques to assess body composition in healthy individuals, mainly in estimating FM, but it has low precision in surgical and oncology patients as it underestimates total body water and FFM despite having a correlation (r^2^) of 0.6275 and 0.274 with DXA and CT, respectively, in estimating FFM [89].

However, it is useful to estimate body composition in several clinical conditions, such as sarcopenia, aging, cancer, and obesity. A higher FM, but also lean mass, estimated by BIA, has been shown to lead to a higher risk of atrial fibrillation [90]. Estimation of FFM and other variables by predictive equations may be inadequate due to errors in acute and critical illness. It is likely that underlying assumptions, such as hydration, individual subjects’ stability, and body geometry, are unreliable in these situations, particularly in obese patients, as BIA errors increase with adiposity [91].

In order to improve body composition assessment, BIA devices can be multifrequency/BIA spectroscopy, segmental, and/or vectorial.

The ratio of extracellular/intracellular water by BIA spectroscopy is a marker of muscle and nutritional health, as a higher ratio relates to lower strength and functionality in the elderly [84]. Despite the improvement in water body composition assessment with multifrequency BIA/spectroscopy, a systematic review evaluated sixteen studies using this technique and found an overestimation of FM in 11/16 and FFM in 9/16 studies. The reliability of the technique was high for groups but not for individuals [92].

Segmental BIA can help to gain measurement precision in skeletal muscle mass assessment and can be used to evaluate fluid distribution and changes in pathological states such as renal failure and ascites [47].

Bioelectrical impedance vectorial analysis (BIVA) phase angle, mainly standardized by sex, age, and BMI, is useful as a prognostic factor as it is lower in acutely ill hospitalized patients and predicts longer hospital stay, nutritional and functional status, morbidity and survival of cancer [89], and critically ill patients [93]. Garcia-Almeida et al. found that a phase angle < 3.95° is a cut-off of mortality risk in acute COVID with 93.8% sensitivity and 66.7% specificity [94]. In other clinical settings, a meta-analysis found that patients with cardiovascular disease had a lower phase angle than a control group [95]. BIVA evaluation of overhydration is useful in fluid management in critically ill patients [96] and predicts mortality, heart failure, and acute kidney failure [91]. Recently, phase angle has shown good accuracy in detecting sarcopenia, with cut-off values of 5.95° and 5.04° for young and old males, respectively, and 5.02° and 4.20° for females [97].

Despite its limitations, BIA is a technique accepted by the Global Leadership Initiative on Malnutrition (GLIM) group to assess muscle mass for the diagnosis of malnutrition [98,99].

The advantages and limitations of BIA and other techniques are shown in Appendix A.

#### 3.3.3. Computed Tomography

The attenuation of X-rays is different depending on the tissues they pass through, and this is the basis of body composition assessment by CT. Pixel density in CT images is expressed in Hounsfield Units (HU), which measure tissue attenuation compared with water (HU = 0) and represent the mean X-ray absorption in the studied area. Predetermined HU scales are used to identify body tissues by quantifying density in muscle mass (HU = −29 to +150), subcutaneous and intramuscular adipose tissue (HU = −190 to −30), and visceral adipose tissue (HU = −150 to −50) [89].

Body adipose tissue and skeletal muscle cross-sectional area (CSA) estimated from a preselected level of a CT image have a good correlation with whole-body composition by DXA. This has been observed in cancer patients, although the validity in acute and critically ill patients is unknown [47,91].

CT is used for assessing muscle quantity through CSA of different muscles, usually psoas or all the muscles present at the L3 level of the Skeletal Muscle Index (SMI). Muscle quality can be estimated by Hounsfield unit average calculation (HUAC) for the muscle tissue [91] or muscle attenuation (MA), as increased fat infiltration in muscle results in lower HU count, though some studies try to assess muscle quality by measuring intermuscular (but not intramuscular) adipose tissue (IMAT) [100]. Visceral and subcutaneous adipose tissues can be assessed in the same image used for muscle evaluation. Non-contrast CT images must be selected as contrast enhances muscle attenuation values [101]. The methodology to analyze CT images using the ImageJ free software from the U.S. National Institutes of Health [102] was published some years ago [103], although the estimation of visceral adipose tissue (VAT) has a coefficient of variation of up to 12.3% [102], and interobserver reliability had an intraclass coefficient of 0.94 [104]. Muscle quantity as SMI is inversely related to age and in-hospital mortality in the elderly [105].

Muscle quality assessed by muscle density (HUAC) in cancer, trauma, elderly, and surgery patients (including waitlist) were more predictive of morbidity and mortality than muscle quantity variables, such as CSA or SMI, although both muscle quantitative and qualitative variables do not correlate well with muscle function [91,105]. Lower muscle density assessed by CT has also been found in women with distal radius fractures and has been associated with more fragility and poorer functional outcomes, but although the quantitative variable CSA was inversely related to fragility, it was not associated with fragility fractures [106].

The advantages and limitations of this technique and others are shown in Appendix A.

#### 3.3.4. Magnetic Resonance Imaging

Magnetic resonance imaging, which does not provide ionizing radiation, is based on the magnetic properties of nuclei of some chemical elements, mainly hydrogen in water and fat, by which a magnetic field makes hydrogen nuclei absorb energy and align them in a known direction. When the magnetic field stops, hydrogen nuclei release energy as they return to the previous state, and this energy is the signal that is used to generate the magnetic resonance image [47,107].

As hydrogen protons in fat and water have different magnetic resonance frequencies, Dixon MRI can distinguish fat and water into two different images, and so, MRI is useful to assess regional adipose tissue and fat infiltration in organs [107]. Furthermore, as subcutaneous adipose tissue has a significantly shorter T1 than visceral adipose tissue, making it possible to distinguish between both, it could be useful to assess brown adipose tissue, as brown adipose tissue contains a lower fat fraction than white adipose tissue [108].

MRI is also useful for the assessment of CSA of skeletal muscles and provides an estimation of body composition and muscle quality disturbances, such as muscle disruption, oedema, fat infiltration, or fibrosis, increasing fat and fibrosis with age and pathologies [97,100]. Quantitative MRI combines signals of fat and lean mass and free water. These signals can be separated as they show different amplitude and relaxation times, but previous calibration with oil, meat, and water is needed [109].

MRI correlation for body composition assessment with ADP was 0.984, with BIA 0.75 and 0.81 for females and males, respectively, and with DXA, 0.99 for body fat (CV 4.5%) and 0.97 for lean tissues (CV 4.6%), showing high correlation with BIA spectroscopy [107,110]. Although quantitative MRI measurement of FM has high accuracy and very low CV (0.44–1.42%), it underestimates FM in adults at higher values up to 15% and overestimates in adults at lower values and children up to 10% [47,110].

Globally, MRI assessment of body fat distribution and characteristics is accurate and reliable, and muscle CSA is smaller in dialysis patients than in controls due to significant muscle atrophy, and these patients had muscle infiltration by water, fat, and/or fibrosis, which can increase muscle volume (but not quality), as has been found in patients with liver cirrhosis due to overhydration of muscle [111]. Quantitative MRI has been shown to be useful in evaluating muscle fat replacement in patients with facioscapulohumeral muscular dystrophy type 1, detecting the progression of disease even before it is evident in strength and functional tests, and results are significantly correlated with clinical diagnosis of obesity, sarcopenia, and sarcopenic obesity [110].

The advantages and limitations of this technique are shown in Appendix A.

#### 3.3.5. Air-Displacement Plethysmography

The ADP measures directly the body mass and body volume through air displacement inside a sealed chamber. From these measurements, body density is computed using the known density of fat and age- and sex-specific FFM density coefficients to determine the total body FM and FFM [76,111].

This is a non-invasive method, particularly convenient in infants because it is relatively rapid to perform and not affected by movements, thus not requiring sedation or immobilization [112]. There are two commercially available ADP systems (Cosmed, Rome, Italy): the Pea Pod, designed for infants up to 6 months (or 8 kg), and the Bod Pod, for children older than 5 years, adolescents, and adults. More recently, an adapted seat for the Bod Pod was made available for children aged 3–5 years. However, it was reported that ADP may be not well tolerated in children aged 1–5 years, thereby limiting its use [111].

The ADP (Pea Pod) was validated in infants against known bovine phantoms [113] and deuterium dilution [114]. In small infants and children aged 2–6 years, the ADP, compared with the four-compartment model, was found to be highly reliable and accurate for determining the %FM [112,115]. Charts are currently available to assess ADP body composition in preterm and term infants from birth to 6 months of age [116]. In older children, adolescents, and adults, ADP (Bod Pod) is a reliable and valid technique in a wide range of subject types, compared with several reference methods [117,118]. In children and adolescents aged 10–18 years, %FM measured by ADP was highly correlated with %FM measured by DXA and, on average, 2.9% lower than DXA measurements [119]. In adults, APD is highly reliable in measuring body FM [120] and body density [121].

Adiposity can be accurately determined by ADP, using as indicators the %FM (given automatically by the equipment) or by fat mass index (FMI) calculated as FM (kg) divided by length or height (m)^2^ [122]. The FMI has been preferred to %FM since %FM has the disadvantage of being a ratio with FM included both in the numerator and in the denominator as a component of body mass [123]. When using the FMI, the length or height must be measured accurately since an inaccurate measurement when squared magnifies the error of the index while losing the ability to differentiate overestimation from underestimation [124].

The advantages and limitations of ADP and other techniques are shown in Appendix A.

#### 3.3.6. Ultrasound Imaging

Ultrasound is a non-expensive, non-invasive, and portable technique without ionizing radiation [101], which has been used in veterinary medicine and food technology for the last two decades for economic reasons [125,126,127], as a greater fat infiltrate in muscle of cattle implies less quality (and less prize) of meat. It has emerged as a useful tool for measuring visceral adipose tissue in humans [128].

Nutritional Ultrasound^®^ [129] can evaluate subcutaneous and abdominal visceral FM and quantity and quality of muscle mass (ectopic fat deposits in muscle) with a technical equipment similar to other areas such as thyroid imaging.

##### Standard Nutritional Ultrasound^®^ to Assess Malnutrition/Sarcopenia

The most common use of US imaging in clinical nutrition is as a tool to assess malnutrition [46,130], including sarcopenia [104].

European Society for Clinical Nutrition and Metabolism (ESPEN) recommends GLIM criteria to diagnose malnutrition. These criteria include not only classical anthropometry items such as weight loss and BMI but also reduced muscle mass measured by emergent validated tools [98]. Initially, only DXA, BIA, CT, or MRI were accepted as validated body composition measuring techniques [98], but in 2022, it was revised to include US imaging as a useful technical approach for diagnosing malnutrition, if available with appropriate expertise and reference values, with preference to anthropometry and physical examination [99].

The most common format for describing US scans is mode B (brightness or greyscale mode), in which the various tissues under the transducer produce a greyscale image that includes different echo intensities, which can be measured to evaluate intramuscular ectopic deposits of fat [129].

Nutritional ultrasound^®^ **muscle measurement** technique standardized by the SARCUS (SARCopenia through UltraSound) study has established a series of measures of patient position, location of anatomical structures, and systematization of standardized measurement cuts. It is recommended to obtain three images to be stored and the average value of the three image measurements recorded to increase the accuracy of the technique [129,131]. In general, when the US technique is employed to assess muscle mass, it is most employed in the right thigh. Leg angle correction may be necessary to center the image of the quadriceps rectus femoris (QRF) muscle.

Nutritional ultrasound^®^ to evaluate muscle mass is a reliable technique as studies have been published reporting strong comparisons with other techniques such as DXA, CT, and MRI [99]. It has been shown to be similar to hydrostatic weighing and superior to anthropometry in estimating FFM in high school wrestlers [132], and CSA of QRF has shown superiority over BIA spectrometry in assessing muscle wasting and risk of protein energy-wasting in patients in hemodialysis [133]. QRF muscle layer thickness has shown to be an independent predictor of 60-day mortality in critical patients and, with CSA, has correlated with screening tools of malnutrition and sarcopenia, being superior to BIVA phase angle [134] and correlated with CT muscle CSA, with a concordance index of 0.66 to 0.77 to predict low muscularity [135]. Quadriceps thickness and CSA showed a correlation with BIA FFM index and dynamometry in patients with chronic obstructive pulmonary disease [136], and muscle thickness, circumference, and CSA (corrected by squared height) showed a positive correlation with BIVA phase angle in obese female patients [137].

In older adults, muscle thickness showed high a correlation with appendicular FFM measured by DXA [138]. It has been proposed as a low-cost objective method for muscle evaluation in nutritional assessments [139,140].

##### Nutritional Ultrasound^®^ to Assess Visceral Adiposity

Visceral adiposity is measured at the midpoint between the xiphoid appendix and the navel on the midline. In the cross-section, the anatomical structures that are visualized are ordered from the most superficial layer (epidermis) to the preperitoneal fat that must be measured to evaluate ectopic fat deposits, although it is difficult to find its profound limit in persons with obesity, which is an important limitation to this technique but can be solved by measuring ectopic fat in other locations.

In patients with DM2, visceral adiposity assessed by US imaging has correlated with visceral adiposity evaluated by CT, and both had been associated with dyslipidemia, IR, inflammatory markers, intima-media thickness at the common carotid artery, and risk for coronary artery disease [141], and, in patients who underwent a coronarography, predicted the existence and severity of coronary artery disease, but subcutaneous fat did not [142].

##### Nutritional Ultrasound^®^ to Assess Muscle Quality, including Ectopic Fat

This technique can assess muscle quality to:evaluate cardiometabolic risk, as fat infiltration of muscle is a kind of ectopic fat deposition, which is related to cardiometabolic and other complications, as shown above;evaluate muscle functionality, which deteriorates in senescence, frailty, and muscle diseases.

Various parameters provide information about muscle quality [129,131,143]: pennation angle, fascicle length, muscle stiffness, contraction potential, muscle microcirculation, and echo intensity. The latter is increased by fat infiltration of muscle (myosteatosis) but also by degenerative changes with age or diseases, and is decreased in myonecrosis. Perimuscular oedema is also decreased in echo intensity, but outside the muscle [129].

Echo intensity has been evaluated subjectively only, being useful to discriminate muscle deterioration in critical patients, by classifying images into four categories: homogeneous hypoechogenic (not present in critical patients but in controls), heterogeneous hypoechogenic, fat infiltration (most common in critical patients), and fasciitis and/or necrosis (in one-third of critical patients in the study) [144].

Nevertheless, quantifying echo intensity gives a non-subjective variable, which allows better discrimination among muscle quality of different patients or the evolution of the same patient in the clinical setting. This quantitative assessment of echo intensity can be made by 2D texture analysis with different available software. Figure 2 and Figure 3 show the process with the United States National Institutes of Health “Image J” [145].

Echo intensity of QRF has demonstrated inter-rater and test–retest reliability in a validation study developed in patients with knee osteoarthritis, independently of how the muscle Region of Interest had been selected and is suggested to evaluate muscle changes [146].

In a study by Young et al. [147], an independent influence of subcutaneous fat thickness on muscle echo intensity was described, and a correction factor was generated to avoid this influence on the quantification of intramuscular fat. Nevertheless, as fat is not the only muscle infiltrate that increases echo intensity, as with fibrosis (collagen), and both imply a deterioration of muscle quality. Echo intensity itself, without correction, provides information about muscle quality, which correlates with clinical outcomes. This study by Young et al. found a strong association between lower limbs FM assessed by MRI and muscle echo intensity by US imaging in healthy volunteers, much stronger after adjusting for subcutaneous fat thickness [147].

Quantitative echo intensity has also shown to be useful in evaluating frailty in an elegant study, which found that echo intensity (without correction for subcutaneous fat thickness) is lower in healthy young adults than in robust elder adults, higher in prefrail elderly, and highest in frail elderly [148]. This finding has been confirmed [140].

The advantages and limitations of US imaging are shown in Appendix A.

#### 3.3.7. Comparison of Techniques to Evaluate Quantity and Quality of Adipose and Muscle Tissues in Patients with PKU

Techniques to evaluate body composition are compared in Appendix A. The most useful ones in the clinical setting are anthropometry, BIA, and US, as CT and MRI are opportunistic when conducted for other reasons but not for body composition assessments, and DXA and ADP are less available [149,150,151,152,153,154,155,156,157,158,159,160,161,162,163,164,165,166,167,168,169,170,171,172,173,174,175,176,177,178,179].

See Appendix A. Advantages and disadvantages of measuring body composition using simple and predictive measurements, two component technique, and electromagnetic technique, and the summary of advantages and disadvantages of methods to assess body composition.

### 3.4. Cardiometabolic Risk in PKU Patients

#### 3.4.1. Why Cardiometabolic Risk May Be Important in PKU Patients

Patients with PKU are expected to have a normal life expectancy associated with newborn screening and better and earlier treatments. The number of adult patients in follow-up is increasing. Adult patients with PKU may have a higher prevalence of overweight, dyslipidaemia, and IR than the general population, but this may be highly variable, according to several confounders. There are common factors contributing to overweight and obesity in PKU and the general population, such as birth weight, parental weight, and physical activity, and disorder-specific factors, such as clinical severity and childhood growth restriction [180]. Even though the recent meta-analysis does not show an increased risk for overweight and obesity in PKU, patients with classical PKU may be at a higher risk [1]. Their food behaviour, energy intake, physical exercise levels, and dietary adherence are all important for weight balance [181].

Since the late 1970s, there has been evidence of increased BMI and a trend of rapid weight gain among children with PKU, although other studies did not find differences with healthy populations [182]. Evidence is more limited in adult patients with PKU.

Excess adiposity increases cardiometabolic risk in PKU. There are no long-term prospective studies or systematic/meta-analysis on body composition, and there are different techniques used to measure body composition and pubertal status. Variable PKU phenotypes make comparison of the data between studies even more difficult. However, in a disorder where nitrogen for body composition and growth is from an artificial source, it is important to monitor lean mass and FM. Long-term associated health problems, such as DM2 and cardiometabolic health [183,184,185], may be linked to an altered body composition because of a dependency on a protein substitute/medical food, even though a disease specific effect cannot be ruled out.

#### 3.4.2. Causes of Potential Increased Cardiometabolic Risk in Patients with PKU

##### Nutrition Therapy: Increased Intake of Carbohydrates and Body Composition

Overweight and obesity has been reported in patients with PKU, mainly females after puberty. However, the prevalence of overweight and obesity in patients with PKU with controlled energy and protein intakes is similar to that of the control population [1].

There are some modifiable lifestyle factors related to overweight, such as total energy, carbohydrates, special low protein foods, natural protein foods, Phe-free protein substitute, fruit and vegetables, food behavior, dietary adherence, and physical activity [180].

Energy expenditure, fat, and carbohydrate oxidation in PKU is similar to that of control populations [181,186]. Carbohydrate, but not fat intake, is higher than in the general population [181,187]. The intake of natural protein is limited with controlled intakes of protein, with or without animal protein-rich natural foods.

Many low-protein special foods are high in carbohydrates. At the same time, they are lower in fiber, and this increased glycemic index and, therefore, glycemic load, could be a causative factor of overweight and obesity, as well as ectopic fat deposition [188]. 

Finally, Phe-free protein substitutes are rich in fast-absorbing high glycemic index sugars or artificial sweeteners to help improve their bitter taste. Due to the concern of overweight, there are formulae that contain less sugar and, to improve the poor taste of free amino acids, some are derived from glycomacropeptide (GMP). It is important to underline that GMP-based formulations contain residual Phe but have a better taste than amino acid supplements. Some evidence shows that GMP formulae do not lead to higher overweight/obesity rates, nor do they change anthropometry or body composition [189,190], and we do not know any studies that support the association between energy content of formulae and overweight. Nevertheless, we should be very critical with all the formulations and avoid excessive energy intake with some products. There are also some concerns regarding the quality of the protein equivalents these formulae deliver and the kinetic absorption properties [191].

##### Nutrition Therapy: Chronic Exposition to Amino Acids

IR and DM2 are mainly a consequence of abnormal glucose metabolism, even though fat and protein may modulate glucose metabolism [59]. Patients with PKU are recommended to have a higher protein equivalent intake in comparison with the WHO safe levels for protein intake [192]. We still lack robust nutritional epidemiology data to understand if patients with PKU ingest significantly higher protein intakes, particularly because long-term adherence to the dietary regimen is sub-optimal [193].

The role of amino acids (especially branched-chain) is to regulate protein synthesis at the translation initiation level through the mammalian target of rapamycin (mTOR) pathway. This leads to protein synthesis and an increase in cell size and, in adipocytes, leptin secretion, tissue morphogenesis, and control of glucose uptake. Amino acids inhibit response to insulin via mTOR, which has a negative feedback on insulin signaling, contributing to IR. Leucine is the most potent activator of the mTOR pathway in adipocytes [194].

Branched-chain amino acid (BCAA) metabolites (ketoacids) and enzymes in their pathway are involved in systemic inflammatory diseases by causing inflammation in adipocytes or immune cells. People with obesity have abnormal catabolism and increased plasma levels of BCAA, linking obesity with IR [195].

Branched-chain α-ketoacid dehydrogenase (BCKD) has a key role in the BCAA oxidative catabolism to acetyl (leucine) and succinyl coenzyme A (isoleucine, valine) [196]. An impairment of this results in less production of acetyl and succinyl coenzymes A, maintaining the mTOR pathway and the activity of glutamate dehydrogenase and increasing production of reactive oxygen species, leading to mitochondrial damage, autophagia, increased systemic inflammation, and cardiovascular disease [195].

It has been suggested that plasma levels of sulphur amino acids methionine and cysteine/cystine correlate with BMI and FM but not lean mass. Insulin resistance also predicts the development of DM2 [196].

Plasma levels of Phe and tyrosine are higher in people with obesity or DM2 versus healthy people. Both amino acids have shown the highest association with IR or risk of development of DM2 among all the metabolites studied in a review [196]. Acquired metabolic complications are more prevalent among patients with poorer metabolic control (see Section Metabolic Disturbances).

##### Nutrition Therapy: How It Modulates Microbiota and Microbioma Modulates Metabolism

Microbiota comprises a collection of trillions of microorganisms (viruses, bacteria, archea, and eukaryotes). Microorganisms are in the skin, saliva, oral mucosa, vaginal mucosa, and conjunctiva, but the majority resides in the gastrointestinal tract, the gut microbiota [197]. They form a unique environment with a characteristic metabolism complementary to the host, constituting a complex ecosystem that has been associated with human health in the last decades.

The gut microbiota is dominated mainly by two phyla, Firmicutes and Bacteroidetes, which represent 85–90% of total microbiota, with Actinobacteria, Proteobacteria, and Verrucomicrobia being less represented. Taxonomically, they are distributed in phyla, classes, orders, families, genera, and species [198]. Despite the classical role of microbiota to break down food components and nutrients, particularly the ones for which the host does not have enzymatic machinery, synthesizing a range of metabolites, the importance of microbiota in human health has been evidenced in the literature [199,200,201]. It has been accepted that a disrupted microbiota (a dysbiotic microbiota), with either reduced commensal bacteria, increased pathogens, or an overall reduced diversity and richness [202], is associated with poor health outcomes.

The microbiota is established even before birth, and it is influenced by the mother, the type of birth, antibiotic and other drug exposure, type of feeding, and solid food introduction and becomes more stable around 2–3 years old [203]. Although it is fairly resilient, the specific compositional features differ among individuals, and it can be altered by both internal and external stimuli. This microbiota plasticity, along with inter-individuality, makes it difficult to determine a “healthy” microbiota, but it also brings the opportunity to use this plasticity to shape the microbiota. By manipulating external factors, the architecture and biological outputs of the microbiota can be orchestrated to improve human health.

Diet has been recognized as a major driver for microbiota changes. Diet-induced changes can be detectable after 24–48 h of manipulation [204]. So, the host dietary pattern will have a profound effect on the microbiota, favoring the growth of species that uses these fuel sources, and this diet-induced microbial structure will have a crucial impact on host physiology and disease process (Figure 4) [205].

Nutrients can directly interact with microorganisms to promote or inhibit their growth, shaping the gut microbial community. Also, diet-derived antigens and compounds can shape the gut microbiota in an indirect fashion by affecting host metabolism and its immune system. Dietary modulation of the gut microbiota composition and function could result in beneficial or detrimental consequences on host health. This could be due to immunomodulatory effects of the modified microbiota, downstream effects on host gene expression, or alterations in the landscape of microbiota-produced metabolites, which might act locally in the gut or systemically in other organs [206].

Several studies have tried to address the effect of this Phe-restricted diet on the microbiota. Timmer et al. [207] showed different microbiota profiles from healthy controls. Patients with PKU on a Phe-restricted diet showed reduced richness compared to healthy controls without specific diets. In agreement, a lower microbiota diversity was found compared to healthy individuals. Patients with PKU presented a higher abundance of Firmicutes, Actinobacteria, and Proteobacteria and a decrease in Bacteroidetes [208].

The effect of GMP on microbiota modulation has not been completely explored in the context of PKU. A study with a small and heterogeneous group of patients showed that GMP intervention may have a prebiotic effect since it promoted the abundance of Agathobacter and Subdoligranulum [209].

A group of adult patients with PKU was compared to non-PKU controls, and differences were found in a total of 65 genera across five phyla [210], but no significant differences in microbial diversity. It emphasized a decrease in *Faecalibacterium* in adult individuals with PKU, which has already been described in children with PKU [211,212]. The importance of these bacteria in the production of short-chain fatty acids, including butyrate, and the potential anti-inflammatory activity is of note. Predicted metabolic pathways were found to be different between individuals with PKU on a Phe-restricted diet and non-PKU individuals. Carbon fixation, carbohydrate fermentation, and N-acetylneuraminate degradation were upregulated. No differences in diversity were found between individuals. In a recent study, also with adult patients with PKU, with one group on an enzyme substitution therapy and without diet restriction and the other group on a Phe-restrictive diet for 3 years, Verrucomicrobia was the only phylum with different abundance between groups. It was increased in the Phe-restrictive diet compared to the enzyme substitution therapy group [213]. Some differences were also found in genera and species abundance, such as an increase in *Akkermansia*, *Prevotella*, and *Lachnobacterium*, which were more abundant in the conventional treatment. Patients with conventional treatment exhibited a greater abundance of pathways involved in the urea cycle metabolism and ketone body metabolism compared to enzyme substitution therapy. That could be a result of the protein dietary sources between groups. These data are very difficult to compare between studies as the analyses are relative to the groups used: a control of healthy individuals is different from a control of patients with PKU treated with enzyme substitution therapy without diet restriction, or even with different nutritional interventions. It is possible the Phe-restricted diet may mask the microbiota characteristics induced by the genetic condition of the disease or by the disease state.

The impacts of different supplements and different nutritional and therapeutic strategies on metabolic outcomes are starting to gain attention, and the microbiota has appeared as a promising variable to explore.

These findings regarding gut microbiota in patients with PKU could explain part of the differences in body composition and cardiometabolic risk through changes in energy homeostasis and food behavior.

#### 3.4.3. Cardiometabolic Risk Factors in Patients with PKU

There are some studies evaluating overweight and/or obesity prevalence in patients with PKU; this is mainly in pediatric patients [127,214,215,216,217,218,219,220,221,222,223,224,225,226,227,228,229,230], but also in mixed pediatric and adult populations [182,191,231,232,233,234] or only adults [181,235,236,237,238] reporting different results. This has been recently reviewed [1,239], and cardiovascular risk through MetS concerns not only increased body weight but also increased body fat and, more precisely, increased visceral and other ectopic fat deposition [59].

##### Body Composition

No studies about visceral fat, myosteatosis, or muscle fat infiltration in patients with PKU were identified. There were also few studies with information about body composition, shown in Table 1.

In pediatric patients with PKU, one study reported higher FM by BIA in females, overweight, and in those patients with lower protein intake [219]. A further study reported higher FM in prepubertal non-adherent patients [217] and in prepubertal than in postpubertal ones [240], but neither of them compared results with a control group.

Some studies compared FM by BIA with a control group or the reference population and found no differences [222,224,241], although one of the studies was developed only during the first year of life and compared with a control group only at birth [224]. One study reported lower FM in PKU vs. controls [215], and another found higher FM among males with PKU vs. controls and also in females but without statistical significance [226].

There were three studies that assessed body composition by DXA in children. One of them reported a %FM of 28% to 35%, but data were not compared with a control group [242]. A second study reported that FM in patients with PKU “was generally close to that expected in healthy children”, but data were not shown [243]. The last one reported no differences in FM vs. a control group, with a significant increase in FM z-score with puberty and a positive correlation between FM and blood Phe levels, which could be explained by low dietary adherence [232].

Body composition was evaluated with body ADP in 20 children with PKU compared with 20 controls, and this reported significantly higher %FM in PKU, 25.2% vs. 18.4% [228].

Fat mass and total body protein were estimated by neutron capture analysis in 37 children with PKU and 27 controls, showing less total body protein in PKU [244]. The same group evaluated resting energy expenditure and total body fat in 30 children with PKU and 65 control children without any difference in the range of body fat between both groups [186].

Five studies reported results of body composition in children and adults. Two of them used BIVA and found a %FM of 25.5% to 28.9% and a phase angle of 6.7–6.8, but none were compared with a control group [189,190]. The other two groups did not find differences in FM assessed by BIA in patients with PKU compared with matched or related controls [185,231,245]. This study was conducted with DXA and found that the mean android to gynoid ratio of distribution of FM in adult patients was above the recommendation in both sexes and that the mean fat mass index was 5.0 in children and 9.1 in adults, but it was not compared with controls [236].

Finally, three studies assessed body composition in adults with PKU. In a Spanish cohort, FM was assessed by BIA, showing a mean %FM of 24.6% (19.3–31%). Thirty-four percent of patients were obese [238]. In a study by Barta et al. using multifrequency BIA, female (but not male) adult patients with PKU had higher FM and lower muscle mass, protein, and mineral content compared with controls [233]. Finally, Alghamdi et al. [181] did not find any statistically significant differences in body composition in ten adults with PKU vs. nine controls, although mean %FM, determined by deuterium diluted in drinking water, was 39.4% in PKU versus 34.3% in controls. They found that overall adherence to amino acid formulae, natural protein exchanges, and diet in general were negatively related to FMI [181].

**Table 1 nutrients-15-05133-t001:** Summary of body composition studies in patients with PKU.

Age Group	Method	Study	Findings
Pediatric/Mixed	BIA	Camatta 2020 (94, Brazil, 10–20, 14.0) [219]	+fat in: ♀, overweight, less protein intake
		Tummolo 2022 (36, Italy, 11.4) [217]	+fat in non-compliant patients
		Tummolo 2021 (30, Italy, 4.1–18, 10.8) [240]	fat in prepubertal > postpubertal
		Evans 2017 (37/21, Australia, 0.6–18, 8.8) [222], Huemer 2007 (34, Austria, 8.7) [224], Dobbelaere 2003 (20, France, 0.6–7) [241]	Fat mass: PKU = general population
		Bushueva 2015 (257, Russia, <18) [221]	Fat mass: PKU < general population
		Sailer 2020 (30/30, USA, 5–18) [226]	Fat mass: PKU > general population
		Pena 2021 (11, Portugal, 15–43, 28) [189], Pinto 2017 (11, Portugal, 27.0) [190],	Fat mass 25.5%−28.9%, phase angle 6.7–6.8
		Rocha 2013 (89/78, Portugal, 14.4) [245], Weng 2020 (22/22, Taiwan, 8–27, 15.2) [231]	Fat mass: PKU ≈ control group (20.74% vs. 18.67%)
	DXA	Daly 2021 (48, UK, 5–15, 11.1) [242]	Fat mass 28−35%
		Adamczyk 2011 (45, Poland, 5–18, 13.8) [243]	Fat mass: PKU ≈ healthy population
		Doulgeraki 2014 (80/50, Greece, 10.88) [232]	Fat mass: PKU = control group; + in high [Phe]
		Jani 2017 (27, USA, 4–50, 16) [236]	FMI: children 5.0, adults 9.1
	Plethysmography	Albersen 2010 (20/20, Netherland, 6–16, 10.0) [228]	Fat mass: PKU 25.2%, control group 18.4%
	Neutron capture anal	Allen 1995 (30/65, Australia, 4–17, 9.6) [186], Allen 1996 (37/27, Australia, 3.9–11, 7.3) [244]	Protein: PKU < control, Fat mass: PKU = control
Adults	BIA	Dios-Fuentes 2022 (90, Spain, 16–56, 29) [238]	Mean fat mass 24.6% (19.3–31%), obesity 34%
		Barta 2022 (50/40, Hungary, 28.97) [235]	Fat mass: PKU ♂ + fat mass,—muscle,—protein and—mineral vs. controls
	Deuterium	Alghamdi 2021 (16/15, UK, >10, 25.5) [181]	Fat mass: PKU 39.4% vs. control group 34.3%

Author year (subjects/controls, age range, median/mean age). FMI: fat mass index.

##### Metabolic Disturbances

There are few studies evaluating cardiometabolic disturbances in patients with PKU (Table 2). In 84 pediatric patients, a study reported that 65.5% had low HDL and 51.9% had high triglyceride levels, and this was more prevalent in overweight patients [217]. A further study found normal mean triglycerides, HDL, or fasting glucose levels but did not show the prevalence of dyslipidemia or impaired fasting glucose and did not assess IR [240]. A third study found HDL was lower, and triglycerides and C-reactive protein were higher among overweight patients. Insulin resistance by HOMA-IR was increased in 64.3% of overweight patients but was as high as 35.7% in non-overweight patients [220]. Finally, one study reported lower HDL and higher triglycerides and basal insulinemia among overweight patients, but mean basal insulinemia was as high as 3.8 μUI/mL among the non-overweight patients [230].

In the articles published with a mixed sample of children and adults with PKU, HDL level was lower and triglycerides, systolic, and diastolic BP were higher in patients with PKU. Homocysteine levels were more remarkable among the overweight patients, with higher insulin, Peptide C, and HOMA-IR and lower QUICK index, and higher among patients with PKU vs. hyperphenylalaninemia and vs. controls, with homocysteine levels correlated with BMI, WC, and age [182,233,234].

Finally, in adults, a study found significantly higher BP components, heart rate, total cholesterol, LDL/HDL ratio, markers of inflammation, and oxidative stress and lower HDL levels in patients with PKU vs. controls. The results were worse in patients with high blood phenylalanine levels [237]. The final study of 90 adult patients reported a high prevalence of cardiovascular risk factors [238].

**Table 2 nutrients-15-05133-t002:** Summary of studies with cardiometabolic disturbances in patients with PKU.

Age Group	Study	Findings
Pediatric/Mixed	de Almeida 2020 (84, Brazil, 2.4–19.9, 10.7) [216]	Low HDL 65.5%, 51.9% hypertriglyceridemia. More prevalent in Ow.
	Tummolo 2022 (36, Italy, 11.4) [217]	Normal fasting glucose, triglycerides, and HDL
	Silveira 2022 (101, Brazil, 10–20, 14.8) [220]	-HDL, +triglycerides, +C-reactive protein in ow. HOMA-IR increased: 64.3% Ow. vs. 35.7% non-Ow.
	Kanufre 2015 (58, Brazil, 4–15, 9.1) [230]	-HDL, +triglycerides, +insulinemia in Ow. Basal insulinemia 3.8 μUI/mL in non-Ow.
	Rocha 2012 (89/79, Portugal, 3–30, 14.4) [182], Couce 2016 (141, Spain, 0.5–50, 15.5) [234], Couce 2018 (83/68, Spain, 4–52, 19.27) [233]	-HDL, +triglycerides, +BP in PKU vs. HPA. +homocysteine (+in ow.), +peptide C, +HOMA-IR,—QUICK index in PKU vs. HPA and control. Correlation with BMI, WC, and age.
Adults	Azabdaftari 2019 (33/28, Germany, 18–47, 30.8) [237]	+BP, +heart rate, +cholesterol (total, LDL/HDL ratio), +inflammation, +oxidative stress, -HDL in PKU vs. control. Worse in poorly controlled.
	Dios-Fuentes 2022 (90, Spain, 16–56, 29) [238]	High BP 7.9%, DM2 2.2%, hypercholesterolemia 15.6%, hypertriglyceridemia 17.8%, hyperhomocysteinemia 18.2%

Author year (subjects/controls, age range, median/mean age) Ow.: overweight; BP: blood pressure; HPA: hyperphenylalaninemia; DM2: type 2 diabetes mellitus.

#### 3.4.4. The Challenges of Analyzing Scientific Evidence in Rare Diseases

In patients with PKU, only a small potential pool is available to be recruited for clinical studies due to the rarity of the condition. This scarce pool of potential participants is further restricted based on PKU genotype differences. PKU is heterogeneous in terms of mutations, which has consequences on biochemical and metabolic phenotypes, therefore introducing variability in terms of treatment needs.

Assessing and retaining patients with PKU is also challenging. Patients with PKU may have functional and neuropsychological disabilities, which represent serious barriers to the implementation of the study protocol, namely adherence to the intervention diet, study schedule visits, and self-monitoring assessments. Patient adherence to study visits is challenging, particularly in adolescents and adults, which can then be aggravated by the duration and frequency of follow-up visits or even by logistical barriers in the accessibility to clinics [246]. Moreover, PKU management still differs widely across countries [192]. being problematic in gathering data from different clinical centers in multicenter studies or meta-analyses. A European guideline was published aiming to standardize the diagnostics and treatment of patients with PKU, assuring the best clinical healthcare practices [192]. This initiative also represents an important step to harmonize clinical research on PKU at the European level, specifically as a critical measure of producing strong scientific evidence from observational studies, which is highly dependent on clinical practices.

Another important aspect is the need for consensus to establish which outcomes and outcome measurement tools are most appropriate in designing clinical studies involving patients with PKU. This will allow the development of large, well-designed, and adequately powered clinical studies. Indeed, in 2020, Pugliese et al. [247] indicated that the outcomes reported in the PKU literature have substantial heterogeneity. Furthermore, the instruments used for outcome measurement were diverse. This heterogeneity in the study endpoints limits the ability to synthesize the evidence produced. Recently, nine pediatric core outcome sets (COSs) and measurement recommendations have been developed and published for PKU clinical trials [248]. Hence, future clinical trials involving children with PKU (≤12 years of age) should use COSs, which will allow comparisons across studies and are needed to synthesize clinical data over time. This harmonization will be critical in providing high-quality evidence in PKU research.

Regarding study design, randomized controlled trials (RCT) are the gold standard for supporting causality statements, especially when well-designed, adequately powered, and double-blinded studies are designed and implemented. However, the design and implementation of RCT may be unethical (for instance, placebo/control groups are not an option for patients that need treatment), or when there are no ethical issues, randomization of rare populations is a high-demanding task for PKU researchers, dependent on multiple recruitment centers. For these reasons, it is not surprising that most clinical studies published in the PKU field are cohorts and cross-sectional and descriptive studies. Moreover, many of these studies have relevant design limitations, namely insufficient information on study design (e.g., participants’ selection criteria and description, sample size justification, methods used for dietary assessments, outcome measures according to exposure levels, and efforts to address potential sources of bias) or lacking statistical power.

Overall, clinical research with patients with PKU is challenging. Nevertheless, several promising methodological approaches have been proposed in the design of observational studies (self-controlled, case-control, and prospective inception cohort designs) [249] and RCTs (adaptive, crossover, registry-based, and early escape designs) [250,251]. Likewise, future research involving preferably international multiple research centers should pursue new strategies to improve recruitment and adherence, particularly in adolescents and adulthood, new biomarkers (for long-term and surrogate endpoints), and COSs (standardized high-quality outcomes and associated outcome measurement instruments). These methodological approaches would lead to an optimization of future studies in PKU and contribute to improve the evidence produced in this field.

## 4. Discussion and Future Perspectives

In PKU, profiling of body composition has a fundamental role in nutritional evaluation and understanding the efficacy of nutritional therapy and pharmaceutical interventions. It should be an integral part of the care pathway. It provides essential information about the mass and structure of tissues and organs, and it is an important indicator of long-term health. The dietary treatment in PKU may alter normal physiologic processes, and it is well established that changes in body composition occur when there is a mismatch between nutrient intake and requirements. Tracking body composition rather than just weight is an essential part of improving nutritional outcomes. Periodic monitoring in an outpatient setting may allow personalized nutritional management strategies based on lean body mass rather than the relatively blunt instrument of body weight [252]. It will also enable the application of early preventative action to avoid the development of obesity and MetS.

Reliable tools for measuring body composition are essential. Standardized measurements should be collected in a systematic manner for longitudinal follow up. There is no agreed or validated method of measuring body composition in PKU, with available body composition measurement methods ranging from simple to complex. The choice of a specific method or combination of methods depends on their accuracy, precision, patient burden, age suitability, staff technical skills, test speed, availability of reference data, cost, and safety considerations such as radiation exposure. All methods have limitations and some degree of measurement error and include assumptions that do not apply to all individuals. The more accurate models commonly use a combination of assessments.

Natural protein quality and amount are important in defining lean body muscle mass. Evans et al. [222] showed in a small group of children with PKU that a natural protein intake > 0.5 g/kg/day was associated with improved body composition. Huemer et al. [224] also showed a relationship between natural protein intake and lean body mass. However, the impact on body composition changes in response to pharmaceutical treatment strategies that aim to increase both the quantity and quality of protein intake in PKU is yet to be determined. For patients with classical PKU, there is reliance on “synthetic” nitrogen sources. Lean mass is dependent on the availability of amino acids, but little is known about the impact of low Phe/Phe free protein substitutes with different sources and amounts of macronutrients on body composition. However, Daly et al. [242] did show a trend towards improved longitudinal growth, a reduction in FM and %FM, and improved lean body mass when children with PKU were given GMP only in comparison to Phe-free amino acids as their source of protein substitute.

Body shape and body composition distribution are strong indicators of metabolic health. Although a recent systematic review and meta-analysis of PKU found that the BMI of patients with PKU was similar to healthy controls, a subgroup of patients with classical PKU had a significantly higher BMI. The authors also noted a trend towards a higher BMI in females with PKU in all studies with male and female datasets [1]. Increased fat mass, including how it is distributed, is considered an important contributor to obesity-related health risks, including DM2 and cardiovascular disease. Increased ectopic fat is an important indicator of metabolic dysregulation, especially of IR and cardiac risk. It is prone to inflammatory infiltration and is, therefore, secreting large quantities of proinflammatory, pro-atherogenic cytokines and free fatty acids [253]. Increased intramuscular adipose tissue or muscle fat infiltration has been associated with reduced mobility and increased risk of DM2 [254].

Age-related change in body composition has varying effects on morbidity and disability. The first population of screened patients with PKU is reaching their mid-fifties. Many with classical have experienced a lifetime history of artificial and imbalanced nutrition, so monitoring body composition changes with aging is important. Body composition in aging is characterized by an increase in FM and a decrease in lean tissues, including skeletal muscle mass, which in older adults is related to reduced muscle strength and functional capability, as well as greater morbidity, particularly in women. Women carry a relatively larger proportion of their body weight as FM and, consequently, have proportionately lower muscle mass compared with men. Obesity and dynapenia (age-related loss of strength) rather than sarcopenia (age-related loss of muscle mass) are more important predictors of physical function in older people [255].

## 5. Conclusions

Despite inconclusive results in the pediatric population, there is evidence showing that overweight with increased FM and metabolic cardiovascular risk factors may constitute an important clinical issue in non-adherent adult patients with PKU. A systematic monitoring of body composition within the nutritional assessment protocol will produce relevant information, aiming to develop nutritional management and hydration strategies to optimize metabolic control of adult patients with PKU while preventing cardiometabolic complications.

## Figures and Tables

**Figure 1 nutrients-15-05133-f001:**
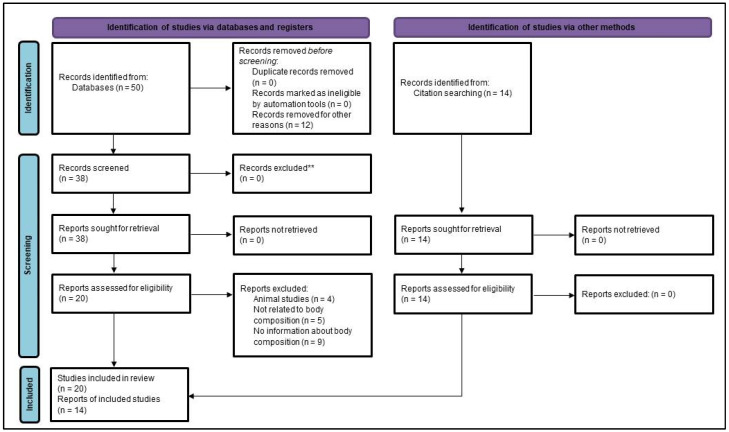
Review process of the main objective: body composition in phenylketonuria.

**Figure 2 nutrients-15-05133-f002:**
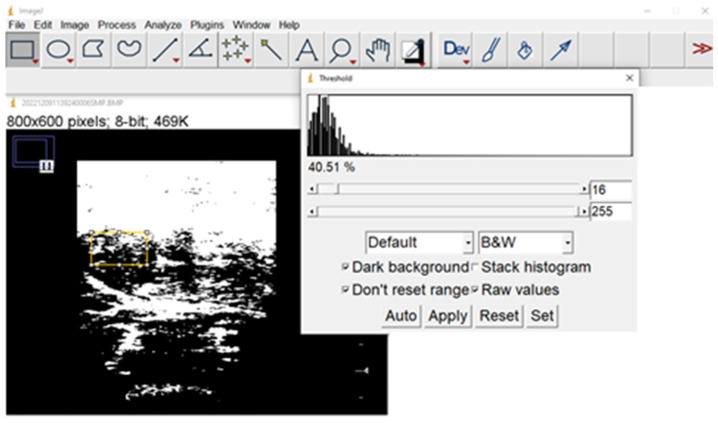
An 8-bit ultrasound image with the selection of one Region of Interest (ROI) and the histogram showing distribution of echo intensities within the ROI.

**Figure 3 nutrients-15-05133-f003:**
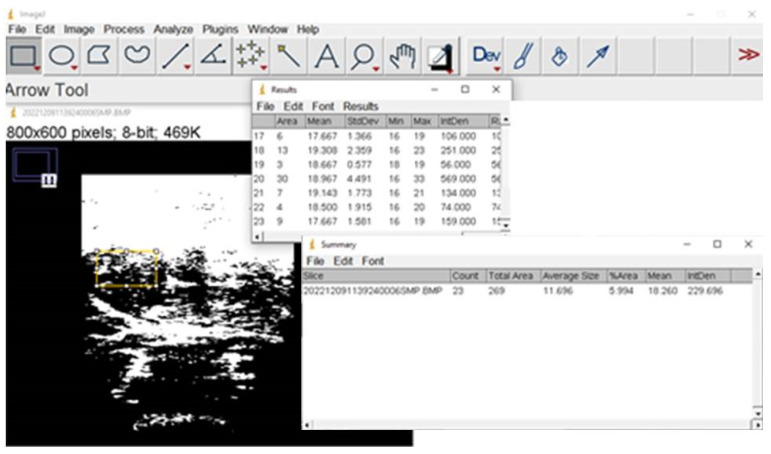
Data table with all the measures and summary of echo intensities in every file within the Region of Interest.

**Figure 4 nutrients-15-05133-f004:**
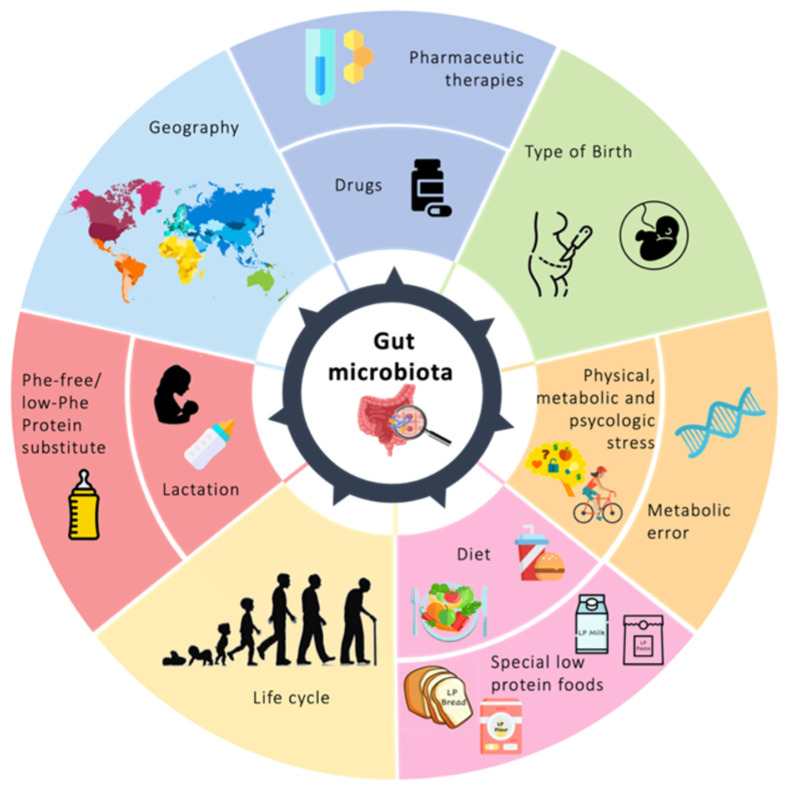
Factors affecting gut microbiota. The inner circle shows the known factors that modulate microbiota. In the outer circle, there are additional factors specific to the PKU population.

## Data Availability

Not applicable.

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
