# Peer review of "Body Composition Evaluation and Clinical Markers of Cardiometabolic Risk in Patients with Phenylketonuria"

_nutrients, 2023, doi:10.3390/nu15245133_

Round 1

Reviewer 1 Report

Comments and Suggestions for Authors

Dear authors,

Thank you very much for your very detailed and thorough investigation, regarding body composition and clinical markers of cardio metabolic risk in patients with PKU.

Here are some suggestions to improve this article:

The title might be partially misleading. Since more than half of the text does not talk about PKU-patients but about markers of cardio metabolic risk, body composition and techniques to evaluate adipose and muscle tissue in general. The reader does not expect such a board overview. Therefore, it should be mentioned in the title.

e.g.: Overview of body composition and clinical markers of cardio metabolic risk in general and in patients with phenylketonuria.

You close your “Introduction” with your primary and secondary objective. However, in the “Material and Methods” section you are only talking about your primary objective. Could you please add the information for your secondary objective?

The following suggestion would be helpful, but not necessary: Could you add three small columns to the Table 1S with effort, risks and costs? You might use signs as “ 0   +   ++   +++”. That would give a simple overview of the possible methods.

There is a typo in the first sentence of the introduction: it has to be “phenylalanine” and not “phenilalanine”.

There is another typo in the first sentence of 3.4.2.2. It has to be “but” and not “bat”.

First sentence on page 15: You state: “Adult patients may have a higher ….” Please add higher than who? As the general population? As children?

Please explain the abbreviation DXA at its first use in the first sentences on page 5 and not at the end of the chapter.

Please add the closing bracket on page 16 after “(a dysbiotic …”

Author Response

Reviewer A.

Dear authors,

Thank you very much for your very detailed and thorough investigation, regarding body composition and clinical markers of cardio metabolic risk in patients with PKU.

Here are some suggestions to improve this article:

  1. The title might be partially misleading. Since more than half of the text does not talk about PKU-patients but about markers of cardio metabolic risk, body composition and techniques to evaluate adipose and muscle tissue in general. The reader does not expect such a board overview. Therefore, it should be mentioned in the title.

e.g.: Overview of body composition and clinical markers of cardio metabolic risk in general and in patients with phenylketonuria.

Thank you very much. We agree with you, the title does not include information for all the content and we need to include the evaluation of body composition. We consider the title will change less and will include this topic by adding a word: “Body composition evaluation and clinical markers of cardiometabolic risk in patients with phenylketonuria”

  1. You close your “Introduction” with your primary and secondary objective. However, in the “Material and Methods” section you are only talking about your primary objective. Could you please add the information for your secondary objective?

We thank the reviewer for this comment. We proceeded accordingly.

  1. The following suggestion would be helpful, but not necessary: Could you add three small columns to the Table 1S with effort, risks and costs? You might use signs as “0   +   ++   +++”. That would give a simple overview of the possible methods.

Thank you very much for the suggestion. Due to the complexity of Table S1, we have decided to add Table S2 with this suggestion.

  1. There is a typo in the first sentence of the introduction: it has to be “phenylalanine” and not “phenilalanine”.

Thank you. We corrected this.

  1. There is another typo in the first sentence of 3.4.2.2. It has to be “but” and not “bat”.

Thank you. We corrected this.

  1. First sentence on page 15: You state: “Adult patients may have a higher ….” Please add higher than who? As the general population? As children?

Thank you. We corrected this.

  1. Please explain the abbreviation DXA at its first use in the first sentences on page 5 and not at the end of the chapter.

Thank you. We corrected this.

  1. Please add the closing bracket on page 16 after “(a dysbiotic …”

Thank you. We corrected this.

Reviewer 2 Report

Comments and Suggestions for Authors

The manuscript provides a detailed description of the methods of assessing nutritional status, cardiometabolic risk factors with special focus on the study of patients with phenylketonuria. The paper is written in accessible language; however, I have some questions and suggestions for the Authors.

1.       There is a lack of implementation of abbreviations for disease states or parameters , which would make it much easier to read and find information in the text. Please search the text for revisions. e.g. in the introduction, it would be useful to provide abbreviations, i.e. MetS (metabolic syndrome), DM (diabetes mellitus), IR (insulinresistance), etc. LDL was introduced without developing the abbreviation in the first place.

2.       Please add correct terminology - LDL is low-density lipoprotein, a fraction, not cholesterol.

3.       Please add a literature reference of exactly which MetS guideline the Authors used.

4.       Last paragraph on the second page - please add in the manuscript an explanation of what the higher risk of metabolic and cardiovascular disorders may be due to.

5.       Did the Authors include all studies on PKU patients in the review or were any rejected, if so, please give in general what were the most common reasons for this. Please add - were papers written in a language other than English also included?

6.       Why did the authors not consider carrying out a systematic review ? What was the reason for combining such two wide-ranging topics into one publication (the part on methodology of nutritional status assessment, etc., and the part on the studies of patients with PKU?)

7.       Chapter two should be accompanied by a selection flowchart to make it easier to observe changes in the elimination of studies for various reasons. The authors could take inspiration from the PRISMA flowchart.

8.       The selection process is described only for studies involving PKU patients, I did not find information on the selection of other publications in Chapters 3.4.1 - 3.4.2.

9.       Section 3.4.3.1. , table 1 and 2 - it is not clear to me to interpret the information from the column "findings", it does not give much information , there is only residual information about what groups were studied, but for most of the rows there are no specific results to compare. It would be well to present the most important results , so that they can be easily compared with other Authors.

Author Response

  1. There is a lack of implementation of abbreviations for disease states or parameters, which would make it much easier to read and find information in the text. Please search the text for revisions. e.g. in the introduction, it would be useful to provide abbreviations, i.e. MetS (metabolic syndrome), DM (diabetes mellitus), IR (insulinresistance), etc. LDL was introduced without developing the abbreviation in the first place.

We thank the referee for highlighting this. We have accepted your suggestions for the most used terms, except if they appear at the beginning of a sentence.

  1. Please add correct terminology - LDL is low-density lipoprotein, a fraction, not cholesterol.

We thank the referee for this remark. We included only as low-density lipoprotein (LDL), as well as HDL, without including the word “cholesterol”.

  1. Please add a literature reference of exactly which MetS guideline the Authors used.

Thank you. In the meantime, during our revision we removed that paragraph, considering there is no need to state here the cut-offs for determining the presence of metabolic syndrome, as they are several and well-known.

  1. Last paragraph on the second page - please add in the manuscript an explanation of what the higher risk of metabolic and cardiovascular disorders may be due to.

Thank you. We have amended this.

  1. Did the Authors include all studies on PKU patients in the review or were any rejected, if so, please give in general what were the most common reasons for this. Please add - were papers written in a language other than English also included?

There were excluded only the articles lacking data on body composition or metabolic parameters. No article was excluded by language reasons if at least it included an English abstract to pass the filter.

  1. Why did the authors not consider carrying out a systematic review? What was the reason for combining such two wide-ranging topics into one publication (the part on methodology of nutritional status assessment, etc., and the part on the studies of patients with PKU?)

Earlier this year, the systematic review from Takeu et. al. (Tankeu, A.T., Pavlidou, D.C., Superti-Furga, A. et al. Overweight and obesity in adult patients with phenylketonuria: a systematic review. Orphanet J Rare Dis  18, 37 (2023). https://doi.org/10.1186/s13023-023-02636-2.) studied excess of body weight in patients with PKU. Our aim here is to update knowledge regarding not only body weight but also body composition and metabolic consequences and, as there is a growing interest in clinicians working with inherited metabolic diseases in the assessment of body composition, the update had to include this. We totally agree with you and now we are planning to do a systematic review following the PRISMA protocol as a next step in our line of research.

  1. Chapter two should be accompanied by a selection flowchart to make it easier to observe changes in the elimination of studies for various reasons. The authors could take inspiration from the PRISMA flowchart.

Thank you, included as Figure 1.

  1. The selection process is described only for studies involving PKU patients, I did not find information on the selection of other publications in Chapters 3.4.1 - 3.4.2.

Thank you. We have included a new paragraph at the end of the Material and Methods section regarding this, as the other reviewer suggested.

  1. Section 3.4.3.1. , table 1 and 2 - it is not clear to me to interpret the information from the column "findings", it does not give much information, there is only residual information about what groups were studied, but for most of the rows there are no specific results to compare. It would be well to present the most important results, so that they can be easily compared with other Authors.

We agree with you. Tables reflects the heterogeneity of design and results and lack of quality of the studies with information about excess of body weight and body composition of patients with PKU and this is why they did not give more information.

Round 2

Reviewer 2 Report

Comments and Suggestions for Authors

The authors responded to my comments and suggestions sufficiently.